# Pauses and Parsing: Testing the Role of Prosodic Chunking in Sentence Processing

**Caoimhe Harrington Stack [1,\*] and Duane G. Watson [2]**

[1] Department of Computer Science, Vanderbilt University, Nashville, TN 37212, USA

[2] Department of Psychology and Human Development, Vanderbilt University, Nashville, TN 37212, USA

[\*] Correspondence: c.stack@vanderbilt.edu

**Abstract:** It is broadly accepted that the prosody of a sentence can influence sentence processing by providing the listener information about the syntax of the sentence. It is less clear what the mechanism is that underlies the transmission of this information. In this paper, we test whether the influence of the prosodic structure on parsing is a result of perceptual breaks such as pauses or whether it is the result of more abstract prosodic elements, such as intonational phrases. In three experiments, we test whether different types of perceptual breaks, e.g., intonational boundaries (Experiment 1), an artificial buzzing sound (Experiment 2), and an isolated pause (Experiment 3), influence syntactic attachment in ambiguous sentences. We find that although full intonational boundaries influence syntactic disambiguation, the artificial buzz and isolated pause do not. These data rule out theories that argue that perceptual breaks indirectly influence grammatical attachment through memory mechanisms, and instead, show that listeners use prosodic breaks themselves as cues to parsing.

**Keywords:** intonational phrase; prosody; syntactic disambiguation; pauses; perceptual breaks





## 1. Introduction

Prosodic phrasing, which includes the timing and phrasing of spoken language through pausing and word lengthening, plays an important role in processing syntactic structure (Steedman 1991; Wagner 2005). For example, the sentence "I met the daughter of the colonel who was on the balcony" is ambiguous, but its interpretation can be disambiguated by the phrasing of the sentence through prosodic boundaries (Carlson et al. 2001). A boundary after the word *daughter*, which would be signaled through the placement of a pause, a drop in F0, and lengthening of the word *daughter*, biases listeners towards a reading in which the colonel is on the balcony. A boundary after the word *colonel* biases listeners towards an interpretation in which the daughter is on the balcony. There is general agreement that boundary placement influences the processing of these types of attachment ambiguities (Lehiste et al. 1976; De Pijper and Sanderman 1994; Price et al. 1991; Swerts 1997; Wightman et al. 1992; Cutler et al. 1997; Frazier et al. 2006; Kraljic and Brennan 2005; Schafer et al. 2000); however, there is disagreement surrounding how prosodic information is used by the sentence processing system in the parsing of syntactic structure. This paper explores whether these processing preferences are a byproduct of how words are prosodically phrased together through perceptual breaks or whether processing is driven by intonational cues to syntactic attachment. More generally, we investigate whether the temporal–perceptual grouping of words influences the grammatical structuring of a sentence.

Listeners can use intonational boundaries to make inferences about the syntactic structure because intonational boundaries regularly occur at syntactic boundaries (see Wagner and Watson 2010 for a review). However, although there is a correlation between syntactic and intonational boundaries, the relationship is not one-to-one (Shattuck-Hufnagel and Turk 1996). Major syntactic boundaries can occur in the absence of prosodic boundaries

and vice versa. This is in part due to the fact that there is a fair amount of optionality in how a speaker might choose to produce any given sentence. Consider (1) below where "//" indicates the presence of an intonational boundary.

(1a)　John adopted a dog during the pandemic.

(1b)　John adopted a dog // during the pandemic.

Both versions of (1) are acceptable even though they differ in where a prosodic break is placed, demonstrating that a given syntactic structure is compatible with multiple prosodic phrasings. Similarly, different syntactic structures can be consistent with similar prosodic phrasing. Consider (2) below:

(2)　　The shooting of the hunters was terrible.

Example (2) is syntactically (and lexically) ambiguous but placing a prosodic boundary at different locations in this sentence has no effect on how listeners interpret it (Lehiste et al. 1976; Shattuck-Hufnagel and Turk 1996). Thus, the prosodic structure of an utterance can provide probabilistic information about the syntactic structure, but not deterministic information, raising the question of how exactly prosodic information is used by listeners to make inferences about sentence processing.

There are two classes of theories that have been proposed to explain prosodic effects on parsing: theories based on prosodic grouping and theories based on interpreting prosodic boundaries as probabilistic cues. We review both below and discuss their predictions.

## 2. Prosodic Grouping

One class of theories proposes that effects of intonational phrasing on parsing are driven primarily by how intonational boundaries group dependent words together (Schafer 1997). The most well-articulated version of this theory is the Visibility Hypothesis (VH) (Frazier and Clifton 1998). Under this theory, boundaries degrade the memory representations of recently encountered words, which reduces their "visibility" to incoming words that must be integrated into the syntactic representations. This means that if an intonational boundary must be crossed when integrating two dependent words, processing difficulty is higher than if the words are in the same intonational phrase. This has consequences for ambiguity resolution: listeners should prefer to attach incoming words to attachment sites in the same intonational phrase over attachment sites that are in a different intonational phrase because this reduces processing complexity. For example, consider the ambiguous sentence in (3):

(3a)　The bus driver stopped // the rider with a mean glare.

(3b)　The bus driver stopped the rider with a mean glare.

(3c)　The bus driver stopped the rider // with a mean glare.

The prepositional phrase (PP) *with the mean glare* could either indicate what *the bus driver* used or describe a characteristic of *the rider*. The VH proposes that intonational phrases encapsulate information into perceptual packages that make information within the phrase more visible than information outside of it. When attachment sites are more visible, the parser can more easily attach incoming words to them. This means that in (3a), when the comprehender encounters the PP and must make a decision about where it attaches, *the rider* is a more visible attachment site than the verb *stopped*. Because *the rider* is contained within the same intonational phrase as the PP, attaching to this site requires fewer resources than if the parser tried to cross over the boundary to attach the PP to *stopped*. In contrast, in (3b), both *the rider* and *stopped* are in the same intonational phrase as the PP, so listeners should find both attachment sites equally visible for attachment. As predicted by the VH, Schafer (1995) found that listeners increase their rate of VP interpretations in 3(b) as compared to 3(a).

Note, however, that grouping theories do have limitations. They cannot explain, for example, why a late boundary in example (3c) creates a preference for high PP attachment to the verb *stopped*. Because both attachment sites are equally inaccessible to the incoming

PP, it is not clear why there would be a preference for the attachment site that is farthest away. We return to this point below.

A second important point is that the mechanism underlying grouping effects is either unspecified or varies from theory to theory. For example, in some models, prosodic boundaries actively trigger processing such that material within an intonational phrase is processed together (e.g., Slowiaczek 1981; Marcus and Hindle 1990). In models such as the Visibility Hypothesis described above, effects of prosodic phrasing are an emergent property of memory limitations of the processing system. Words that are farther away from each other are more difficult to integrate because they require keeping the initial dependent in memory. Frazier and Clifton (1998) argue that prosodic phrasing can increase the distance between dependents, but so can syntactic complexity, pitch accenting, visual segmentation, or even just time. This characterization of visibility suggests that a wide range of factors, including acoustic disruptions such as an extraneous noise occurring between two dependents, could potentially decrease the visibility of a recent dependent. We return to this point below.

In sum, prosodic grouping theories argue that intonational phrases serve as processing units, and as such, integrating words within these chunks is easier than integrating words across chunks. In these theories, attachment preferences are a byproduct of how intonational phrases package words together and the decay in memory representations over time.

### 3. Prosodic Cues

A second class of theories proposes that, rather than serving as processing units, the prosodic structure provides junctures that listeners use as probabilistic cues to differing syntactic structures. The idea that listeners use statistical models of the linguistic structure to make inferences and predictions in sentence processing is not new, and in fact, is pervasive in psychological models of linguistics processing. Listeners use probabilistic cues to make inferences about phonological categories (e.g., Kleinschmidt and Jaeger 2015), the syntactic structure (e.g., Levy 2008), and even the prosodic structure (e.g., Kurumada and Roettger 2022).

This idea was formalized in Watson and Gibson (2005) as the Anti-Attachment Hypothesis (AAH), which states that listeners disprefer attaching an incoming phrase to an attachment site that is followed by an intonational boundary. This claim is rooted in the distribution of intonational boundaries in language production. Speakers tend to produce intonational boundaries at syntactic constituent boundaries, particularly if those syntactic constituents are long (Watson and Gibson 2005). If listeners are sensitive to the distribution of speakers' preferences for intonational boundary placement (see a similar claim in MacDonald 2013), then boundaries could potentially serve as a probabilistic cue to syntactic closure, which should lead to a dispreference for attaching to words that precede a boundary.

If we consider example (3) again, the AAH makes different predictions than prosodic grouping theories. In (3a), the boundary is predicted to encourage attaching *with a mean glare* to *rider* because the verb *stopped* is followed by a boundary. Similarly, in (3c), the boundary after *rider* should signal that it is dispreferred for attachment and so listeners should prefer to attach to the verb *stopped*.

An additional prediction of cue-based theories is that the presence of the cue itself, i.e., intonational boundaries, is critical for pushing around listeners' syntactic preferences. Whereas grouping theories rely on timing such that words that are chunked together are more easily integrated with one another, cue-based theories are predicated on the idea that the boundary itself is an important cue.

In the three experiments below, we test this prediction. All three experiments involve examining structures such as (4):

(4)    Susie learned that Bill (a) telephoned (b) after John visited.

In (4), *after John visited* could attach to *learned* or to *telephoned*. Based on the previous literature (e.g., Wagner and Watson 2010), we expect that intonational boundaries at (a) and (b) should influence listeners' interpretations. A boundary at (a) should bias listeners toward a reading in which John visited after the telephoning event. A boundary at (b) should bias listeners towards an interpretation where Susie learned about the event after John visited.

In Experiment 1, we replicate the finding that boundaries at (a) and (b) in example (4) bias interpretation in the expected ways. In Experiments 2 and 3, we maintain the prosodic chunking of these critical stimuli, but replace the intonational boundaries with a buzzer sound in Experiment 2 and a pause with no pre-boundary word lengthening in Experiment 3.

This manipulation allows us to test versions of prosodic grouping theories, such as the Visibility Hypothesis, in which multiple constraints conspire to influence the visibility of recently produced words (Frazier and Clifton 1998). Frazier and Clifton (1998) argue that visibility is an emergent property of a memory system in which representations decay over time. Increases in visibility can be caused by a wide range of factors, including just time. Thus, we test Frazier and Clifton's (1998) Visibility Hypothesis in which properties of the acoustic signal that increase the temporal distance between dependents will decrease the visibility of the initial dependent. The prediction is that prosodic chunking should still shift around syntactic preferences, even if an acoustic stimulus that is not a traditional prosodic boundary occurs at the edges of these prosodic chunks. Cue-based theories predict that in the absence of an intonational boundary cue, listeners' syntactic preferences will not be influenced by breaks such as noises and non-intonational boundary-generated pauses.

## 4. Experiment 1

The goal of Experiment 1 was to validate the paradigm. In this study, we first test whether intonational boundaries influence syntactic attachment as predicted by AAH and VH theories. If so, we can compare the performance to the artificially constructed perceptual breaks in Experiments 2 and 3.

### 4.1. Methods

Participants. Subjects were recruited from Amazon Mechanical Turk and paid USD 4.00 for completion of the experiment. Thirty subjects completed the experiment and were retained for the analysis. This study was approved by the university's institutional review board.

Stimuli. Twenty-eight critical items were adapted from Carlson et al. (2001). Critical items had PPs that were ambiguous as to where they should be attached, as in (4) above. Each of the 28 critical items was recorded twice, once with an early boundary and once with a late boundary. The average acoustic measurements of the stimuli can be seen in Table 1, where a comparison is made between sentences where the boundaries occurred and the measurements taken in the same place when the boundary did not occur. In addition, 42 filler sentences that did not contain a PP ambiguity were recorded. All the critical and filler items were recorded by a female native English speaker with a Midwestern U.S. accent.

Forced-choice comprehension questions, which gave subjects two answer choices, were created for each item. Comprehension questions for critical items always probed whether subjects had interpreted the sentence as low- or high-attachment. For example, after hearing the critical sentence "Susie learned that Bill telephoned after John visited", subjects were asked the following:

What happened after John visited?

1.   Susie learned something after John visited.
2.   Bill telephoned after John visited.

**Table 1.** Mean values of selected acoustic cues in the control sentences. Early vs. Late refers to the placement of the main boundary in the sentence. Boundary vs. Non-Boundary refers to whether measurements were taken from the sentence where the boundary appeared in that location, or from the alternative sentence where a boundary did not appear in that location.

|  | Early: Boundary | Early: Non-Boundary | Late: Boundary | Late: Non-Boundary |
|---|---|---|---|---|
| Duration of Pre-Boundary Word | 480 ms | 380 ms | 410 ms | 330 ms |
| Fundamental Frequency of Pre-Boundary Word | 176.74 Hz | 198.61 Hz | 156.17 Hz | 165.25 Hz |
| Pause Duration after Pre-Boundary Word | 200 ms | ~0 ms | 190 ms | 20 ms |

In the above example, the first answer represents the high-attachment interpretation and the second choice represents the low-attachment interpretation. Comprehension questions for filler items always had a correct and incorrect option to choose from.

Procedure. The critical and filler items were combined to make four lists of 28 critical items and 42 filler items. Each list was counterbalanced for the boundary location and answer presentation. In each list, subjects heard half of the critical items with an early boundary and half with a late boundary. The presentation of answer choices to each comprehension question was counterbalanced so that half of the low-attachment interpretation answers occurred first, and half of high-attachment interpretation answers occurred first. Likewise, filler item answers were counterbalanced so that half of the correct answers appeared first and half of the incorrect answers appeared first. Within each list, the presentation of filler and critical items was randomized for each subject. In total, each subject heard 70 sentences and answered a comprehension question after each one. Subjects did not receive feedback about whether the question was answered correctly or not.

### 4.2. Results

Subject responses for critical items were coded based on whether their answer indicated low- or high-attachment interpretations and whether the boundary occurred in an early or late position in the sentence. The results indicated that sentences with an early boundary were interpreted as low-attachment 78.57% of the time. Sentences with a late boundary were interpreted as low-attachment 59.05% of the time. These results are displayed in Figure 1. To analyze these results, a multi-level logistic regression was constructed, which analyzed the effect of the boundary location on syntactic interpretation. We included all random effects for subjects and items, but due to problems with model convergence, we iteratively removed random effects that accounted for the least amount of variance until we found a model that converged. The resulting model included random intercepts for both the subject and item, and random slopes for the subject. There was a main effect of the boundary location ($z = -4.334$, $p < 0.001$) indicating that subjects were influenced by where a boundary occurred when making syntactic interpretation decisions. Subjects made significantly more low-attachment interpretations when boundaries were in an early position.

As Experiments 2 and 3 revealed null results, we completed Bayesian analyses in all three experiments. We completed a Bayesian analysis of the multi-level logistical regression that analyzed the effect of the boundary location on syntactic attachment while estimating the random intercepts for both the subject and item and the random slopes for the subject. The estimate was $-1.18$, with a credibility interval of $[-1.77, -0.61]$. As this interval does not cross zero, we reject the null hypothesis in this Bayesian analysis.

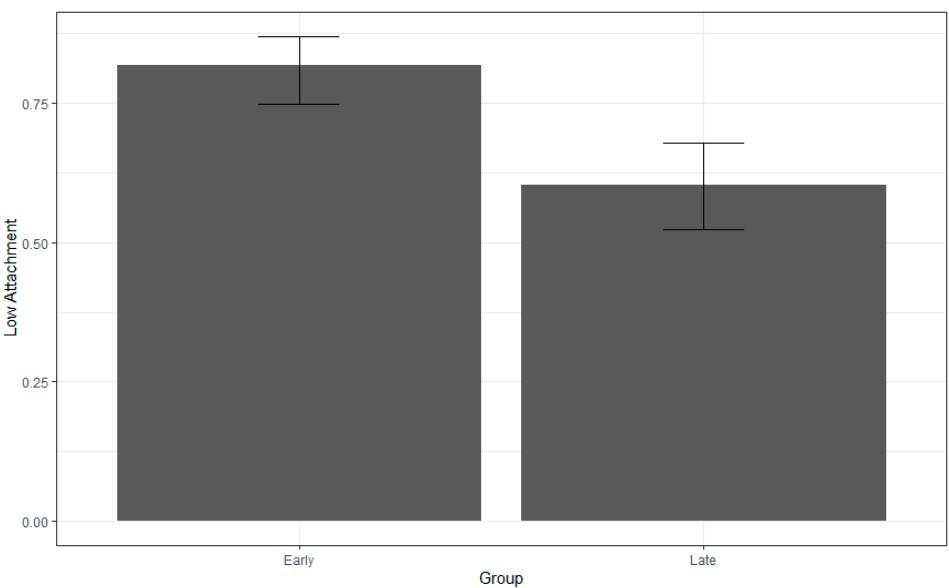

**Figure 1.** Proportion of low-attachment interpretations by boundary location in Experiment 1.

*4.3. Discussion*

These data suggest that the critical items chosen for this experiment were consistent with the predictions of the VH and the AAH. Subjects were more likely to make low-attachment interpretations than high-attachment interpretations when there was an early boundary. Given that these stimuli produced the effects predicted by the VH and AAH, we can use these results as a benchmark against which to compare the results of Experiments 2 and 3.

If the Visibility Hypothesis is correct and sentence analysis is influenced by perceptual units, then altering the sentences from Experiment 1 so that the perceptual units are created in using artificially generated perceptual breaks should result in the same pattern of sentence analysis by listeners. That is, when listeners encounter a perceptual break early in the sentence, the resulting perceptual grouping should bias listeners towards low-attachment because the ambiguous constituent is forced into the same perceptual group as the low-attachment site. Regardless of how the perceptual breaks are produced, the VH predicts that the results will mirror the pattern of responses seen in Experiment 1. To test this, Experiments 2 and 3 removed the naturally produced prosodic boundaries by cross-splicing the early part of the late boundary sentence with the late part of the early boundary sentence and replaced them with artificially created break cues—in Experiment 2, a basketball buzzer, and in Experiment 3, a second-long pause.

**5. Experiment 2**

In Experiment 2, sentences were edited so that prosodic boundaries were removed and a non-linguistic cue, a basketball buzzer, was placed to overlap with speech where the boundary would have started. This resulted in sentences with a clear auditory cue appearing in the place of the prosodic boundaries, though overlapping with the speech. The goal of this was to create perceptual units that were bounded by non-linguistic information. If the boundary–syntax relationship is the result of how information is distributed across perceptual units, the interpretation of ambiguous sentences should be identical no matter what cue is used to create perceptual units. However, if listeners fail to use the non-linguistic buzzing sound to make interpretation decisions, it will suggest that the boundary–syntax relationship is not due to visibility constraints on the parser.

### 5.1. Methods

Participants. Subjects were recruited from Amazon Mechanical Turk and paid USD 4.00 for completion of the experiment. Forty subjects completed the experiment, but one was excluded for indicating that English was not their native language, resulting in 39 subjects being retained for the analysis.

Stimuli. The twenty-eight critical items adapted from Carlson et al. (2001) and used in Experiment 1 were used again. The goal of this experiment was to create the same perceptual units produced by the prosodic boundaries in Experiment 1 but with an entirely new cue. Taking the naturally produced sentences with early and late boundaries from Experiment 1, they were edited so that the acoustic cues of the boundary were removed and replaced with a non-linguistic sound. To remove the natural boundaries, the sentences were edited so that the first half of sentence 4(b), i.e., "Susie learned that Bill", was spliced to the second half of sentence 4(a), i.e., "telephoned after John visited" so as to create a natural sounding sentence with the major prosodic boundaries removed. Then, a basketball buzzer noise, with a duration of 330 ms and no silence around it, was inserted where the boundary had originally occurred to create perceptual units of speech separated by a non-prosodic cue. The audio of the sentence continued while the buzzer was played, resulting in the basketball buzzer overlapping with speech. This resulted in sentences that were interrupted partway through with the sound of the buzzer. See https://doi.org/10.17605/OSF.IO/DVSW3 to hear the stimuli. Two sentences were created for each critical item so that each item had one sentence with an early buzzer and one with a late buzzer. In addition, the 42 filler sentences without the PP ambiguity were edited so that the buzzer occurred at random points. The same forced-choice comprehension questions from Experiment 1 were used for both critical and filler items.

Procedure. The sentences were combined to make four lists as in Experiment 1. In each list, subjects heard half of the critical items with an early buzzer and half with a late buzzer. The answers to each comprehension question were counterbalanced so that half of the low-attachment interpretation answers occurred first, and half of high-attachment interpretation answers occurred first. Filler item answers were counterbalanced so half of the correct answers appeared first and half of the incorrect answers appeared first. Within each list, the presentation of filler and critical items was randomized for each subject. In total, each subject listened to 70 sentences and answered a comprehension question after each one. Subjects did not receive feedback about whether the question was answered correctly or not.

### 5.2. Results

Subject responses were coded based on whether their answer indicated low- or high-attachment interpretations and whether the buzzer occurred in an early or late position. The results indicated that sentences with an early buzzer were interpreted as low-attachment 68.13% of the time. Sentences with a late buzzer were interpreted as low-attachment 69.96% of the time. These results can be seen in Figure 2. To analyze these results, a multi-level logistic regression was constructed, which analyzed the effect of the boundary location on syntactic interpretation. We included all random effects for subjects and items, but due to problems with model convergence, we iteratively removed random effects that accounted for the least amount of variance until we found a model that converged. The resulting model included random intercepts for both the subject and item. There was no effect of the buzzer location ($z = 0.818$, $p = 0.41$), indicating that subjects did not alter their rates of low-attachment interpretations based on where a buzzer occurred within a sentence.

As the results for this experiment failed to reject the null hypothesis, we completed a Bayesian analysis to investigate how much evidence there was for the null result. We completed a Bayesian analysis of the multi-level logistical regression that analyzed the effect of the boundary location on syntactic attachment while estimating the random intercept for each subject. The intercept estimate was 0.12, with a credibility interval of $[-0.17, 0.41]$. As this interval does cross zero, we accept the null hypothesis in this Bayesian analysis.

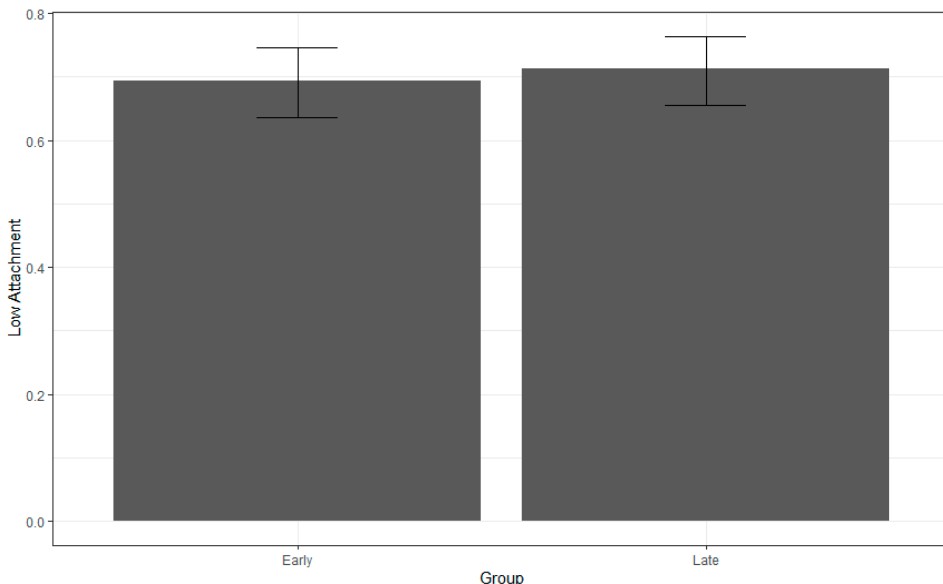

**Figure 2.** Proportion of low-attachment interpretations by buzzer location in Experiment 2.

*5.3. Discussion*

　　The data showed that subjects did not use the perceptual units created by the buzzer to make interpretation decisions about the sentence. Subjects who heard the late buzzer were just as likely to make low-attachment interpretations as subjects who heard the early buzzer. Subjects showed a bias for low-attachment interpretations of critical sentences regardless of boundary placement. This suggests that the interpretation of ambiguous sentences is not influenced by processing constraints that focus the syntactic analysis within perceptual units.

　　The results of this experiment are inconsistent with the VH. If sentence processing is influenced by how words are perceptually grouped, then comprehenders should continue to be influenced by perceptual breaks regardless of how the breaks are created. Because subjects were not sensitive to processing units created by the buzzer, it suggests that processing units are not driving syntactic attachment.

　　However, there are three potential explanations for why the buzzers failed to influence syntactic attachment. First, the buzzers may have distracted the listeners. Perhaps the parser generally *does* focus its processing within perceptual units regardless of how these units are created, but particularly distracting acoustic information diminishes this tendency. If sentence interpretation is influenced by constraints on processing resources, it may be the case that these processing resources are consumed by particularly odd acoustic cues. It would not matter what attachment sites are more or less visible to the parser if the parser does not have the resources to analyze them properly. If this is the case, it would explain why subjects showed a low-attachment bias across all sentences. Native English speakers have a low-attachment bias overall (Carreiras and Clifton 1999) and listeners may have just reverted to this bias when overwhelmed with a distracting cue.

　　It may also be the case that the buzzers did not create the perceptual units they were intended to make. The purpose of the buzzer was to create perceptual units. In Experiment 2, the buzzer sound occurred *over* the spoken audio where the boundary would have occurred. It may be that subjects were able to filter out the noise of the buzzer. After all, it is common in everyday conversations for listeners to focus on their partner's speech while ignoring the sound of a typing keyboard, music in the background, a barking dog, or any other number of unrelated sounds. In these cases, listeners may completely separate non-linguistic audio cues from the linguistic signal. Research on the auditory stream analysis may provide an alternative explanation of Experiment 2. The auditory system is thought to use both bottom-up cues and top-down knowledge to integrate and stream

multiple sounds (Bregman 2008). Listeners in Experiment 2 may have been able to use both as a way of focusing on the incoming speech as one perceptual unit, rather than as two perceptual units separated by a buzzer. As a bottom-up strategy, the auditory system has a bias to group similar fundamental frequencies together as a single sound, and treat distinctly different fundamentals as a different sound (Bregman 2008). In addition, sounds that begin at a later time are less likely to be considered part of an already ongoing sound. In the case of Experiment 2, both of these expectations were violated by having a distinctly different audio cue occur partway through the speech. In addition, listeners have a lifetime of experience with the acoustic cues associated with language, but very little (if any) experience with buzzers occurring in a communicative context. This experience with speech might allow listeners to focus on the speech stream and treat the buzzer as a distraction (Hartmann and Johnson 1991). If the auditory system is able to easily categorize a distinctively different sound, such as the basketball buzzer, as separate from the speech stream, the parser may simply ignore the presence of the intrusion in the speech stream. The null results of Experiment 2 may be due to both bottom-up and top-down analyses of the speech and buzzer audio being employed by listeners.

Lastly, the buzzer was overlayed over the speech stream so there was no perceptual break. Thus, it is possible that the buzzer and the speech were perceptually segregated, preventing the perception of a perceptual break. If the buzzer did not impact the timing of the words, this may not have been a fair test of the Visibility Hypothesis. We designed Experiment 3 with stimuli that definitively interrupt the speech stream, removing this potential confound. In Experiment 3, the buzzer that played over the sentences was replaced with an inserted silence. This inserted silence was linguistic-like, in that pauses do occur in conversation, but was also a true break in the speech that could potentially create perceptual units for listeners. If the stimuli in Experiment 2 were perceived as occurring *over* the sentence, the pause in Experiment 3 stimuli now presented a clear break *within* the sentence that created two clearly separate units.

## 6. Experiment 3

The goal of Experiment 3 was to provide a more natural, though still non-linguistic, cue to break sentences into perceptual units. Experiment 3 was thus designed to be similar to Experiment 2, but the buzzer was replaced with a less distracting cue that was more likely to be integrated with the sentence. The buzzer was replaced with a silence that was inserted where the boundary would have occurred. The use of added silence as a cue had the benefit of overlapping with a property of prosodic boundaries (that is, a salient pause) while still being non-linguistic.

If the VH is correct, listeners should use the perceptual units created by the silence to parse sentences. Along with the results from Experiment 1 and 2, this would suggest that sentence processing relies on the visibility of attachment sites, but that this visibility may be ignored when the parser is faced with confusing or difficult acoustic loads. However, if listeners still fail to use these perceptual units for sentence processing, it would suggest that the role that boundaries play in syntactic processing is more than just providing perceptual breaks for the listener. Since the Visibility Hypothesis does not provide an explanation for why only perceptual units created by prosodic boundaries would be considered by the parser, it would suggest that a different language model is needed to explain the boundary–syntax relationship.

A secondary goal of this experiment was to better understand how the perceptual system categorizes and processes intonational boundaries. As we discussed above, intonational boundaries are perceptual units that correlate with a number of differing acoustic features, such as pre-boundary word lengthening, a pause, and a change in F0 (e.g., Shattuck-Hufnagel and Turk 1996). In this study, the break consists of only a pause, which means that there is only partial evidence for an intonational boundary. Pauses can occur in speech without signaling the presence of a boundary. Ferreira and Karimi (2015) make the analogy to playing an instrument: sometimes a musician momentarily stops playing

because a rest is present, and this is a property of the rhythmic structure of a piece. Other times, musicians may stop playing because they do not know what to play next or have made an error. In other words, they become disfluent. In speech, the former case is analogous to an intonational boundary, with a pause as one of several acoustic markers, while the latter is analogous to a pause that signals a disfluency. A key prediction of the prosodic cue-based theories is that it is the presence of an intonational boundary in particular that drives syntactic attachment. It is possible that if we see attachment effects that are driven by the presence of a pause, listeners are interpreting pauses as intonational boundaries, which suggests that pauses are sufficient for the perception of an intonational boundary.

In sum, if we see a difference between the early and late pause condition, it would suggest that either the pauses are changing the relative distance between the possible attachment sites and the ambiguous constituents, or that pauses by themselves can be interpreted as cues to attachment. However, if we see no difference between the conditions, it would suggest that intonational phrase boundaries, specifically the gestalt percept of a boundary created by multiple sources of acoustic information, are necessary cues for attachment.

*6.1. Methods*

Subjects. Forty subjects were recruited from Amazon Mechanical Turk and paid USD 4.00 for completing the experiment. One subject was excluded from the analysis for indicating that they were not a native speaker of English. This resulted in 39 subjects being retained for the analysis.

Stimuli. The stimuli for this experiment were similar to those from Experiments 1 and 2. The same 28 critical items were used, but edited so that instead of boundaries or buzzers, silences created the perceptual units. As in Experiment 2, the naturally recorded items with an early and late boundary were spliced together to create natural sounding sentences with major prosodic boundary cues removed. Then, each sentence had a silence of 1.015 s inserted into the early or late boundary location. A long pause was used to ensure that the inserted silence was salient because, unlike intonational boundaries, which are signaled by multiple acoustic cues, a pause was the only acoustic signal that a perceptual break was present.

The use of an inserted silence to create perceptual units gave subjects a cue that was similar to prosodic boundaries, which are sometimes indicated by a pause in the speech stream. However, the other markers of a prosodic boundary, such as pitch and durational changes, were absent at these breaks. In addition, the use of a computer-generated silence sounded markedly different from a human-produced pause. The resulting effect was that the silence created a cue that had some similarities to prosodic boundaries but was still clearly non-linguistic, like the buzzer. However, the silence was not a jarring cue within the middle of the speech stimuli, making it less distracting than the buzzer. The 42 filler items had a silence of the same length inserted in between two words. The same comprehension questions for critical and filler items from Experiments 1 and 2 were used in Experiment 3.

Procedure. The critical and filler sentences were combined to make four lists similar to Experiment 2. In each list, subjects heard half of the critical items with an early silence and half with a late silence. The answers to each comprehension question were counterbalanced so that half of the low-attachment interpretation answers occurred first, and half of high-attachment interpretation answers occurred first. Filler item answers were counterbalanced so that half of the correct answers appeared first, and half of the incorrect answers appeared first. Within each list, the presentation of filler and critical items was randomized for each subject. Subjects listened to each of the 70 sentences and answered a forced-choice comprehension question.

*6.2. Results*

Subject answers were coded based on whether they made low- or high-attachment interpretations to critical sentences. The data were analyzed to compare the proportion of

high-attachment interpretations made when the silence occurred in the early position as compared to when the silence occurred in the late position. Sentences with an early pause were interpreted as low-attachment 70.13% of the time while sentences with late pauses were interpreted as low-attachment 73.81% of the time. Results can be seen in Figure 3. To analyze these results, a multi-level logistic regression was constructed, which analyzed the effect of the boundary location on syntactic interpretation. We included all random effects for subjects and items, but due to problems with model convergence, we iteratively removed random effects that accounted for the least amount of variance until we found a model that converged. The resulting model included random intercepts for both the subject and item, and random slopes for the subject.

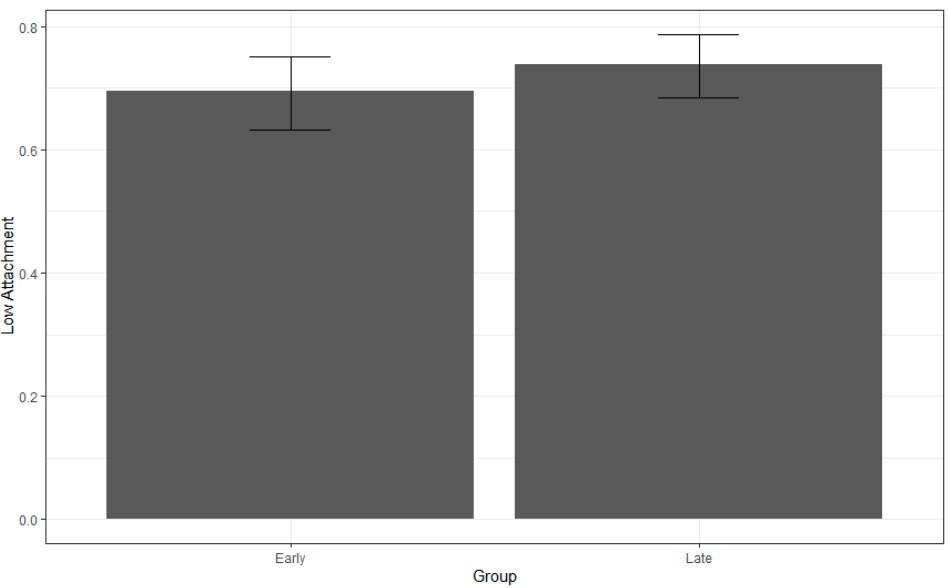

**Figure 3.** Proportion of low-attachment interpretations by pause location in Experiment 3.

The results failed to find an effect of the pause location on sentence interpretation ($z = 1.626$, $p = 0.10$). This indicated that subjects' sentence interpretation was not biased by the location of a pause in a sentence. Regardless of where the pause occurred in a sentence, subjects were just as likely to make a low-attachment interpretation.

As the results for this experiment failed to reject the null hypothesis, we completed a Bayesian analysis to investigate how much evidence there was for the null result. We completed a Bayesian analysis of the multi-level logistical regression that analyzed the effect of the boundary location on syntactic attachment while estimating the random intercept for each subject. The intercept estimate was 0.28, with a credibility interval of $[-0.06, 0.63]$. As this interval does cross zero, we accept the null hypothesis in this Bayesian analysis.

### 6.3. Discussion

Subjects did not use the perceptual units created by computer-generated silences to make decisions about syntactic attachment. Regardless of whether the silence appeared in the early or late position, subjects were equally likely to say that the sentence had a low-attachment interpretation. This indifference to the location of the pause suggests that the Visibility Hypothesis does not appropriately explain the boundary–syntax relationship. If processing is driven by processing resources and the visibility of linguistic information within perceptual units, then sentence analysis should still be influenced by perceptual units created in novel ways. These data suggest that processing resources and visibility alone cannot explain the boundary–syntax relationship.

These data also suggest that the presence of a pause is not sufficient for signaling the percept of an intonational boundary. Otherwise, we would have seen attachment

preferences differ across the two conditions. Instead, it appears as though listeners must hear a combination of cues to perceive an intonational boundary.

Experiment 3 addressed the issues present in Experiment 2's stimuli and still failed to find evidence that listeners were processing sentences based on perceptual units. This suggests that the constraints proposed by the VH on sentence processing do not provide a complete picture of how boundaries and the syntax interact. It may be the case that the foundation of what the VH proposes is correct, but the details are inaccurate. Some potential explanations are considered in the General Discussion below.

## 7. General Discussion

Experiment 1 replicated previous findings showing evidence of the strong relationship between the syntax and boundary placement. This relationship has been shown to be robust in the literature (Shattuck-Hufnagel and Turk 1996; Wagner and Watson 2010), even though boundaries and the syntax do not have a 1:1 relationship. Further, Experiment 1 confirmed that the stimuli chosen were valid for testing the claims of the Visibility Hypothesis. The VH proposes that natural constraints in processing are what drive the syntax–boundary relationship. However, the data presented here are not consistent with the VH: Experiments 2 and 3 introduced non-prosodic perceptual breaks in the place of the naturally produced boundaries. In both cases, subjects' syntactic preferences did not change even though the perceptual grouping of the stimulus was manipulated through the creation of artificial breaks. This suggests that visibility of sentence information due to how perceptual units are packaged is not what drives syntactic attachment.

Although we found an effect of the condition manipulation in Experiment 1, but not 2 or 3, we did not directly compare the experimental effects. To do so, we ran a post-hoc mixed effects model that compared the boundary effect in all three experiments. The resulting model included the interaction between the condition and boundary as well as the main effects of the condition and boundary. Experiment 1 was coded as the baseline. First, we see that there are significant differences between Experiment 1 and the other two Experiments overall. Critically, as shown in Table 2, there is significant experimentation by boundary interactions such that the boundary effect in Experiment 1 differs from Experiment 2 and Experiment 3. Contrasts for this model were also conducted to test the effect of the boundary within each experiment. The conducted contrasts evaluated the effect of the early and late boundary position in each of the three experimental conditions (i.e., control, buzzer, and pause) in the post-hoc mixed effects model. The contrasts show that the control condition's (Experiment 1) proportion of low responses varied significantly more across early and late boundaries ($p < 0.0001$) than the buzzer ($p = 0.6261$) and pause conditions ($p = 0.1337$) (Experiments 2 and 3, respectively). The interpretation of this model's results is similar to our previous, though separate, analyses, which showed that there was a significant effect in the control condition, but not in the buzzer or pause condition.

**Table 2.** Main effects and interaction effects of post-hoc mixed effects model comparing all three experiments.

| **Fixed Effects:** | | | | |
|---|---|---|---|---|
| | **Estimate** | **Std. Error** | **z Value** | **Pr (>\|z\|)** |
| (Intercept) | 1.5976 | 0.2578 | 6.197 | $5.76 \times 10^{-10}$ *** |
| Late Boundary | −1.0733 | 0.2054 | −5.225 | $1.74 \times 10^{-7}$ *** |
| Buzzer Condition | −0.6483 | 0.2744 | −2.363 | 0.0181 * |
| Pause Condition | −0.6702 | 0.2741 | −2.445 | 0.0145 * |
| Late Boundary: Buzzer Condition | 1.1559 | 0.2637 | 4.382 | $1.17 \times 10^{-5}$ *** |
| Late Boundary: Pause Condition | 1.3304 | 0.2648 | 5.025 | $5.05 \times 10^{-7}$ *** |

*, *** indicates significance beyond the 0.001 level.

One potential objection to this work is that effects of perceptual grouping may not be robust to disruption from artificial stimuli. Although we tried to mitigate the disruptive effect of the artificial break stimuli in Experiment 3 by using a non-linguistic pause, it might be the case that this still distracted listeners in such a way that the normal memory processes that are involved in parsing were not brought to bear. It is worth noting that claims about visibility have been argued to apply robustly to both written and auditory sentence processing (Frazier and Clifton 1998), suggesting that visibility effects are not a ephemeral process. The failure of alternatively created units to induce these effects in Experiments 2 and 3 suggests that *if* visibility is the driving factor in syntactic interpretation, then boundaries are a necessary component of visibility in spoken language. This is to say that *if* the VH is correct and visibility is a real driving force in syntactic processing, there is something special about boundaries and their ability to influence visibility in spoken language that needs to be specified. This would suggest that these effects are specifically linguistically driven in nature, rather than due to a general memory constraint. One could envision a version of visibility in which boundaries are specific contributors to the distance such that the system first detects boundaries, and then these boundaries have visibility effects on syntactic attachment. Such an architecture is possible, although such a design would not be able to explain the myriad other non-prosodic factors that are part of visibility effects (Frazier and Clifton 1998). This change would also incorporate a key feature of cue-based theories, i.e., centering the importance of perceiving intonational boundaries in syntactic processing, making the two types of theories close to indistinguishable.

Another potential objection to this research is that, while Experiments 2 and 3 included cross-spliced sentences to remove the natural boundaries, Experiment 1 only contained naturally produced boundaries. Cross-splicing was not used to produce these stimuli. It might be argued that Experiment 1 stimuli were treated differently only because they were not created using cross-splicing. We think this is unlikely because the stimuli in Experiments 2 and 3 were designed to minimize perceptual artifacts that might arise from splicing. The stimuli are available at https://doi.org/10.17605/OSF.IO/DVSW3. However, we cannot completely rule out this potential confound, and future studies will need to investigate this question.

The results of the above experiments are more consistent with theories that intonational boundaries themselves play a key part in driving syntactic attachment preferences. As described in the AAH above, intonational boundaries are treated as probabilistic cues not to attach syntactic constituents to the material that precedes the intonational boundary. Note that this can explain the data patterns from the three studies. When a boundary is present as it was in Experiment 1, it provides a signal to syntactic attachment. When other types of perceptual breaks, such as a pause or buzzer, are present, they are not informative because they do not provide the same type of probabilistic information.

The question of why intonational boundaries serve as useful probabilistic cues to the structure remains an open question. One possibility is that these effects are driven purely by the statistics of language, i.e., a lifetime of experience with English teaches listeners that there is a probabilistic link between the location of intonational boundaries and the syntactic structure. This prior experience with prosody is integrated with other linguistic information such as the frequency, plausibility, and context to make inferences about the syntactic structure. Another explanation may be that the intonational boundaries provide information about the speaker intent. That is, listeners may have metalinguistic knowledge about how boundaries are generated in language production, and know that speakers might be using boundaries to disambiguate sentences (e.g., Snedeker and Trueswell 2003). In either case, the presence of an actual intonational boundary is critical for driving syntactic parsing effects.

## 8. Conclusions

The goal of these experiments was to better understand the relationship that exists between the syntax and prosodic boundaries, specifically focusing on investigating whether

the Visibility Hypothesis accurately explains the link between the two. We confirm that intonational boundaries influence syntactic attachment, but when we alter the acoustic form of the perceptual break in an unnatural way, effects on syntactic attachment disappear, which is inconsistent with the Visibility Hypothesis. This suggests that the role of boundaries in syntactic processing is not due to just processing constraints related to how sentences are perceptually grouped. Rather, listeners use intonational boundaries, specifically the gestalt percept of a boundary with several acoustic cues, as specific probabilistic cues to the syntactic structure.

**Author Contributions:** Conceptualization, C.H.S. and D.G.W.; methodology, C.H.S. and D.G.W.; formal analysis, C.H.S.; investigation, C.H.S.; resources, D.G.W.; data curation, C.H.S.; writing—original draft preparation, C.H.S.; writing—review and editing, C.H.S. and D.G.W.; visualization, C.H.S.; funding acquisition, D.G.W. All authors have read and agreed to the published version of the manuscript.

**Funding:** This research received no external funding.

**Institutional Review Board Statement:** This study was conducted in accordance with the Declaration of Helsinki and approved by the Institutional Review Board of Vanderbilt University (protocol code #160700, approved 08/01/2017).

**Informed Consent Statement:** Informed consent was obtained from all subjects involved in this study.

**Data Availability Statement:** The stimuli, data, and analyses are available for download at https://doi.org/10.17605/OSF.IO/DVSW3 (accessed on 16 May 2023).

**Conflicts of Interest:** The authors declare no conflict of interest.

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
