# Peer review of "Pauses and Parsing: Testing the Role of Prosodic Chunking in Sentence Processing"

_languages, doi:10.3390/languages8030157_

Round 1

Author Response

We thank you for the opportunity to resubmit a revised version of our manuscript. We have taken the reviewer’s comment under careful consideration to improve our paper. The manuscript has been revised to take their feedback into account. Below, please see a list of our responses to the reviewers.

Reviewer 1

Issue 1: Does the AV hypothesis specify what kinds of cues in a signal result in perceptual chunking? The interpretation of the results of this study as providing evidence against at least one form of the AV hypothesis hinges powerfully on the claim that any cues to a chunk boundary should result in lesser visibility/availability of the already-chunked words as a potential attachment sites for later words. Thus it would be useful to have some further discussion, in the introduction, of any such claim in e.g. Frazier and Clifton 1997 etc.

Frazier and Clifton (1998) characterize visibility as a general, gradient, memory-based property of the processing system that can be influenced by a number of different factors, including whether or not a given attachment syntactic attachment site is accented, preceded by a boundary, it’s grammatical location, the grammatical complexity of material following the site, visual breaks in reading, and even just time.  Given the broad nature of the linguistic phenomena that Frazier and Clifton describe as influencing visibility (e.g. right branching sentences, pitch accents, intonational boundaries, simple time based measures, etc...), we think it is reasonable to assume that manipulations like the ones we introduce in Experiment 2 and 3 would also influence visibility.  Shafer’s (1997) prosodic visibility is similar.  Schafer states that “material is continually integrated into the partial phrase structure, allowing higher-level processing decisions to be initiated and, presumably, minimizing memory load,” (p.45), suggesting that visibility effects are the result of a passive memory decay process. Of course, other chunking models describe prosodic boundaries playing a more active role in parsing, where boundaries create processing units for strings that are combined at later stages (Slowiaczek, 1981; Marcus & Hindle, 1990).

One thing to note is that these models were really designed and tested to better understand how prosodic structure influences syntactic parsing, not to understand what acoustic features contribute to prosodic representations, and consequently, visibility effects.  These theories start with the assumptions of prosodic phonological theories of the time, I.e. that there are basic prosodic categories like intonational phrase boundaries that the sentence processing system uses to parse structure, and so they are agnostic about how other acoustic factors, such as a buzzing sound or silence might be processed by the system.

We now contextualize our experiment with the above in mind in the Introduction, explaining that we are testing a) a general version of visibility in which acoustic disruption increases the distance between two related attachment sites  and b) in Experiment 3 we explore the more general question of what counts as an intonational boundary (see our response below).

The key takeaway from the present study is that a generalized notion of visibility where simple distance between dependents makes attachment more difficult cannot explain the effects here.

Issue 2: Relatedly, is there any evidence that the two manipulations carried out in Expts 2 (buzz overlay) and 3 (silence insertion) actually resulted in perceptual chunking? The interpretation presented in the paper would be greatly strengthened if there were either a) independent evidence from other publications indicating that buzz overlap or silence insertion resulted in some kind of chunking behavior, or b) a control condition in this paper showing that either or both of these manipulations resulted in some evidence of chunking, independent of the question of how they influence syntactic parsing. In the absence of such evidence, it is challenging to argue (as is already noted in the discussion) that the result presented here show that perceptual chunking alone is not enough to influence syntactic parsing.

The reviewer raises a good point.  We think the additional text that we describe above will help to address this issue.  We have a changed the introduction to better contextualize our work and now frame the paper that makes it clear that we are testing a very specific version of the Chunking Hypothesis, I.e. that any kind of distance between dependents increases the difficulty of attachment. These changes are on pg. 6.

Issue 3: Can the author(s) expand on the suggestion (in the General Discussion) that there is a more complex relationship between cues to (perceptual) boundaries and syntactic parsing than is acknowledged by a model in which the cues in the signal directly influence parsing? That is, the discussion hints at a model in which there are two steps to this process: first, the cues in the signal are interpreted as evidence for the prosodic (not merely intonational, although intonation is a powerful cue) structure of the utterance, and that prosodic structure is then in turn interpreted as evidence for the syntactic structure. Such a model would integrate the claims of both the VH and AAH models, in the sense that a perceived prosodic constituent (such an Intonational Phrase) might be shipped off to a different processor for interpretation, once it is identified using prosodic boundary cues, rendering its word ‘less visible’ i.e. less accessible as attachment sites for upcoming words.

In the General Discussion, we take the Reviewer’s advice by elaborating on the relationship between boundaries and syntactic parsing. While we are hesitant to describe a staged model (constraint based and interactive theories describe frameworks where distinct stages are somewhat fuzzy), we describe a theory in which 1) intonational boundaries, evidenced by a gestalt of different acoustic features, are used actively by the system as a cue to syntactic structure and 2) this could potentially leads to systems in which boundaries are used as direct cues to attachment or in which memory might play a role.  The discussion is on page 27.

Issue 4: Can the authors consider framing at least a part of the presentation in terms of the cues in the signal to chunks and/or boundaries? That is, one interpretation of the results of Expts 2 and 3 is that neither the buzz overlay nor the silence insertion provide sufficient cues to convince a listener that the speaker intended a prosodic boundary at that location. This is consistent with the discussion currently in the paper. But it might be useful to frame the discussion in terms of what cues are required in the signal to ensure that the listener perceives a boundary at a particular location. Speakers sometimes pause in the middle of a prosodic constituent (e.g. an Intonational Phrase) for reasons that have nothing to do with signaling prosodic structure---e.g. word-finding difficulty, structure-planning difficulty, distraction from the environment, etc.---and a listener may well be able to interpret this pattern of pausing without other boundary cues as evidence for something other than a prosodic boundary. Discussing the results in terms of the number and nature of boundary cues provided by the overlapping buzz and inserted silence manipulations, in contrast to the number and nature of boundary cues provided by speakers signaling a prosodic boundary, might make the picture clearer to a reader.

We agree with this point made by this reviewer and Reviewer 2, and now incorporate the suggested framing into Experiment 3. (pg. 20)

Reviewer 2 Report

This is a nice study investigating whether the effect of prosodic boundaries on syntactic attachment in parsing is purely due to the effect of time on interpretation, or whether prosodic boundaries more directly provide cues for  syntactic attachment. According to the timing hypothesis, prosodic boundaries exert their effect based on the delay they cause before upcoming constituents are available, favoring interpretations under which the previous material was analyzed as a constituent (whether for semantic or syntactic reasons). The main result of the paper is that contrary to this idea, the effect a prosodic boundary depends on the presence of acoustic cues for a juncture, not just on the delay. When the timing delay is covered up by a non-speech sound or by silence, the high attachment effect of the delay is not observed.

I think the paper is close to publishable, but there are three issues that I think need to be addressed before it can be accepted.

1. The stimuli need to be published in a reliable, permanent way, either along with the journal publication, or on OSF or some such site.  It is not possible to fully evaluate this study without access to the perception stimuli, especially the ones with the silence manipulation. I am surprised that Languages does not require including the stimuli of perception studies already at the stage of submission for the review process.

2. The stats that are used to test the hypotheses are not sufficient. There should be another model, fitting the data of all three experiments at once. The crucial prediction for this model is a significant interaction between boundary manipulation (prosodic boundary vs. siren vs. silence) and the early vs. late boundary placement on perceived attachment. This predictor could be dummy-coded, to compare the siren and silence manipulation to the prosodic cue case. It seems extremely likely that this interaction will come out, so I don't anticipate that this will cause any problems. The issue with the current analysis is that the argument relies on interpreting null results in experiment 2 and 3, which is problematic for two reasons:(i) Even the Baysian analysis used here to assess whether the null result in study 2 and 3 can be interpreted leaves room for debate how much we can rely on this assessment. This is a complex statistical question, and assessing it requires more statistical expertise than can be expected from the average reader of Languages. If instead, we saw the evidence in favor of an interaction, we could be much more confident in the conclusions, given that we can put a number of likelihood that the data from what it could plausibly look like under the null hypothesis (which predicts no interaction). And (ii), the hypothesis doesn't actually necessarily predict a null effect, it just predicts a smaller effect. For example, when listeners hear a Basketball siren or a silence that doesn't sound like a prosodic boundary, they probably try to restore the covered signal (similar to the phoneme restoration task). Given the total duration of the relevant part of the signal, they may well be somewhat more likely than not to restore a prosodic boundary. In fact, there is a small trend at least in experiment 3 in this direction. But if this effect had been significant, it would not take away from the conclusions: If there was a sizable interaction such that prosodic cues lead to a much bigger effect on attachment preference, we would still conclude that more than timing is at play.  So even if a future, more powerful study found that there is in fact an effect even in replications of experiment 2 and 3, the conclusions could be still be valid, if there is also a sizable interaction effect. So the null effect is ultimately not crucial to the argument.  And therefore looking at the interaction seems more adequate given the logic of the hypotheses that are being tested, and more relevant in light of future tests of the interpretation of the data advanced in this paper.

3. The motivation of Experiment 3 seems a little inconsistent: On the one hand, silence is used because silence is also a cue to prosodic boundaries, and thus more natural than superimposing a siren. On the other hand, it is critical that listeners do not interpret it as a prosodic boundary, since then we should see an effect on attachment. The results suggest that listeners interpreted the silence as not part of the linguistic signal, but as just superimposed, but this was not directly tested (one could have asked listeners about this in a norming study, or something like this). I think the discussion should make this issue a bit clearer---the results are still interesting, but in a way, this study could have gone either way. If we had seen an attachment effect, we would have concluded that the silence had been 'compatible enough' with it being due to a prosodic boundary. I think what we learn from experiment is that silence is only be treated as part of the signal if the prosody is compatible with there being a boundary. In this context I want to reiterate that readers will really want to be able to listen to these stimuli, in order to better understand this result, so including the stimuli in the publication seems essential.

Overall, this is a nice paper and well worth publishing, but I think these three issues should be addressed before it can be published

Author Response

We thank you for the opportunity to resubmit a revised version of our manuscript. We have taken the reviewer’s comment under careful consideration to improve our paper. The manuscript has been revised to take their feedback into account. Below, please see a list of our responses to the reviewers.

Reviewer 2

  1. The stimuli need to be published in a reliable, permanent way, either along with the journal publication, or on OSF or some such site. It is not possible to fully evaluate this study without access to the perception stimuli, especially the ones with the silence manipulation. I am surprised that Languages does not require including the stimuli of perception studies already at the stage of submission for the review process.

We are happy to provide the experimental materials.  The stimuli, along with the data and analysis scripts,  will be available on OSF, at this link https://doi.org/10.17605/OSF.IO/DVSW3, which is also provided after the conclusion section of the paper.

  1. The stats that are used to test the hypotheses are not sufficient. There should be another model, fitting the data of all three experiments at once. The crucial prediction for this model is a significant interaction between boundary manipulation (prosodic boundary vs. siren vs. silence) and the early vs. late boundary placement on perceived attachment. This predictor could be dummy-coded, to compare the siren and silence manipulation to the prosodic cue case. It seems extremely likely that this interaction will come out, so I don't anticipate that this will cause any problems. The issue with the current analysis is that the argument relies on interpreting null results in experiment 2 and 3, which is problematic for two reasons:(i) Even the Baysian analysis used here to assess whether the null result in study 2 and 3 can be interpreted leaves room for debate how much we can rely on this assessment. This is a complex statistical question, and assessing it requires more statistical expertise than can be expected from the average reader of Languages. If instead, we saw the evidence in favor of an interaction, we could be much more confident in the conclusions, given that we can put a number of likelihood that the data from what it could plausibly look like under the null hypothesis (which predicts no interaction). And (ii), the hypothesis doesn't actually necessarily predict a null effect, it just predicts a smaller effect. For example, when listeners hear a Basketball siren or a silence that doesn't sound like a prosodic boundary, they probably try to restore the covered signal (similar to the phoneme restoration task). Given the total duration of the relevant part of the signal, they may well be somewhat more likely than not to restore a prosodic boundary. In fact, there is a small trend at least in experiment 3 in this direction. But if this effect had been significant, it would not take away from the conclusions: If there was a sizable interaction such that prosodic cues lead to a much bigger effect on attachment preference, we would still conclude that more than timing is at play. So even if a future, more powerful study found that there is in fact an effect even in replications of experiment 2 and 3, the conclusions could be still be valid, if there is also a sizable interaction effect. So the null effect is ultimately not crucial to the argument. And therefore looking at the interaction seems more adequate given the logic of the hypotheses that are being tested, and more relevant in light of future tests of the interpretation of the data advanced in this paper.

We have now included this analysis to General Discussion section of the paper. We find that the interaction model results match our current results. That is, there is a significant interaction between condition (Experiments) and boundary location. Once the contrasts are viewed, it is revealed that the control condition is significant, while the buzzer and pause conditions are not. This is discussed in more detail on page 25.

  1. The motivation of Experiment 3 seems a little inconsistent: On the one hand, silence is used because silence is also a cue to prosodic boundaries, and thus more natural than superimposing a siren. On the other hand, it is critical that listeners do not interpret it as a prosodic boundary, since then we should see an effect on attachment. The results suggest that listeners interpreted the silence as not part of the linguistic signal, but as just superimposed, but this was not directly tested (one could have asked listeners about this in a norming study, or something like this). I think the discussion should make this issue a bit clearer---the results are still interesting, but in a way, this study could have gone either way. If we had seen an attachment effect, we would have concluded that the silence had been 'compatible enough' with it being due to a prosodic boundary. I think what we learn from experiment is that silence is only be treated as part of the signal if the prosody is compatible with there being a boundary. In this context I want to reiterate that readers will really want to be able to listen to these stimuli, in order to better understand this result, so including the stimuli in the publication seems essential.

We have included the suggested framing to the introduction of Experiment 3 on pg. 20. A final line on page 28 provides a link to the stimuli, data, and analyses.

We hope that these changes are sufficient to meet the reviewer’s concerns. We feel that they have improved our paper and made it ready for publication.

Round 2

Reviewer 1 Report

I will not have time to re-review this ms. in the near future, but reading over the author’s response letter, it seems that they have addressed my major concerns. I have no objection to accepting the ms. for publication

Author Response

Thank you. We have submitted some brief changes to address Reviewer 2's concerns in this round. 

Reviewer 2 Report

Thanks for publishing the stimuli, data and code, this makes it much easier to understand the study, and will increase its value to the field and its shelf life!

From the description in the first version, I had not fully understood the manipulation of the experiment 2 and 3, and I think this needs to be clarified in the text still in this version. The following is not a complete enough description of what was done:

p.9
"To test this, Experiments 2 and 3 removed the naturally produced prosodic boundaries and..."

It looks to me that the way the boundaries were 'removed' was by cross-splicing the early part of the utterance with a late boundary before the buzzer/silence, and the late part of the utterance with an early bounary after the buzzer/silence. This is a valid way of doing it, but raising some minor concerns whether the actualy boundary conditions were treated differently simply because they weren't cross-splicied (as opposed to the buzzer/silence stimuli). Listening to the stimuli i dont think this is an issue, but it should be nevertheless be mentioned as a potential confound. It cross-spliciing studies, the standard procedure is that one cross-splicies *all* stimuli, in this case this would mean to cross-splice even in the case of the actualy boundaries by taking two parts from utterances recorded with the same boundary condition, but it sounds like this wasn't done here. I don't think this confound was driving the effect, but it's necessary to acknowledge it.

With respect to the buzzer manipulation:

-- It seems that the buzzer was superimposed on the signal without adding time to the signal in the position whether the buzzer occurs (and the overall sitmuli are shorter than either stimuli with actual boundaries). Perceptually, I think it is very clearly perceived as a separate sound effect (or seperate streamin Bregman's terms) that covers up part of the speech signal. Crucially, it doesn't look like the relative timing between the constituents is affected at all (it's just that part of a word needs to be reconstructed, similar to a phoneme restoration experiment). This needs to be made more explicit in the text, especially because it's entirely different from the manipulation in the pause condition

-- Since the buzzer doesn't change the relative timing of any of the words and constituents, I'm not sure how these stimuli actually serve to test the hypothesis. To the extent that listeners are able to restore the speech signal underneath the signal, wouldn't the visibility hypothesis predict no effect of the buzzer on parsing? I don't this these stimuli really show evidence against the visibility hypothesis.

-- The silence manipulation was done differently, in that a second of slience was added at the splice point. These stimuli thus directly test the  hypothesis whether a time delay affects attachment (predicted by the visibility hypothesis, at least as it is presented here). I think the argument should be based on these stimuli then, because to it seems that only these stimuli serve to make the case? The buzzer stimuli are in a way a good point of comparison for the silence manipulation

-- The silence stimuli kind of sound like an artificial interruption in the speech stream happened which would be impossible unless speech is reproduced with a device. It doesn't sound like a natural chunking of speech (I guess if it did sound like chunking then people WOULD take the silence to give information about chunking). I think it's a striking result well worth publishing that there is no attachment effect, but it does raise some questions what listeners actually do with the silence information here---i guess they explain it away as a device malfunction?  At any rate, if the effect of actual prosodic boundaries was a pure function of the additional time their realization incurs, then the silence manipulation should have had a similar effect, so that part of the arguments and stands.

And some additional comments:

The summary of the results based on the interaction data is inaccurate:

"The interpretation of this model’s results are identical to our previ-534 ous, though separate, analyses showing that there was a significant effect in the Control condition, but not in the Buzzer or Pause condition."

The big model (of all three studies) with the interaction *doesn't* actually show that the effect in the buzzer/silence manipulations were not significant--it shows more relevantly that the effect of the boundary manipulation on the attachment decision was statistically significant **bigger* when it was done with a prosodic boundary than when it was done with a buzzer/silence. For the non-significance, one needs to resort to the individual models. This significance of this interaction is however the way to statistically sure that the apparent differences observed in the individual are really statistically different. (which was also assessed based on the bayes measures, but to me at least the interaction is a more compelling way of testing this).

The logic of the argument is also based on this interaction, not based on the non-significance: Suppose there had been a tiny but significant effect on attachment for the silence manipulation---the conclusions would not change, as long as there had been a much bigger effect for actual prosodic boundaries.

line 325 'the null' > 'the null hypothesis'

Author Response

Please see the attached letter to the editors where Reviewer 2's comments are addressed. Thank you for your help in improving the paper. 
